# HistoCartography: A Toolkit for Graph Analytics in Digital Pathology

\*\*\*\*\*\*\*\*\*\*\*\*\*\*\*\*\*\*\*\*\*\*\*\*\*\*\*\*\*\*\*\*\*\*\*\*\*\*\*\*\*\*\*\*\*\*\*\*\*\*\*\*\*\*\*\*\*\*\*\*\*\*\*\*\*\*\*\*\*\*\*\*\*\*\*\*\*
\*\*\*\*\*\*\*\*\*\*\*\*\*\*\*\*\*\*\*\*\*\*\*\*\*\*\*\*\*\*\*\*\*\*\*\*\*\*\*\*\*\*\*\*\*\*\*\*\*\*\*\*\*\*\*\*\*\*\*\*\*\*\*\*\*\*\*\*\*\*\*\*\*\*\*\*\*

## Abstract

Advances in entity-graph based analysis of histopathology images have brought in a new paradigm to describe tissue composition, and learn the tissue structure-to-function relationship. Entity-graphs offer flexible and scalable representations to characterize tissue organization, while allowing the incorporation of prior pathological knowledge to further support model interpretability and explainability. However, entity-graph analysis requires prerequisites for image-to-graph translation and knowledge of state-of-the-art machine learning algorithms applied to graph-structured data, which can potentially hinder their adoption. In this work, we aim to alleviate these issues by developing HISTOCARTOGRAPHY, a standardized python API with necessary preprocessing, machine learning and explainability tools to facilitate graph-analytics in computational pathology. Further, we have benchmarked the computational time and performance on multiple datasets across different imaging types and histopathology tasks to highlight the applicability of the API for building computational pathology workflows.

**Keywords:** Graph Representation Learning, Computational Pathology, Python API

## 1. Introduction

Recent advancements in tissue-slide digitization have paved way for enhancing storage, sharing capabilities, and computer-aided inspection by leveraging Deep Learning (DL). Most DL approaches analyze tissue images in three steps, namely patch generation, patch-level feature extraction, and feature aggregation to produce image-level embeddings for downstream pathology tasks. However, they suffer from several limitations, (i) the trade-off between operational resolution and adequate context per-patch, (ii) the aggregation is often sub-optimal, (iii) comprehensive modeling of tissue composition is missing, and (iv) the lack of model transparency raises barriers to deployment in real life.

To circumvent these limitations, entity-graphs are proposed (Demir et al., 2004) where the nodes and edges of the graphs denote tissue entities and their interactions, respectively. Entity-graphs, followed by Graph Neural Networks (GNNs)-based processing, have recently gained popularity in addressing various pathology tasks (Zhou et al., 2019; Chen et al., 2020; Pati et al., 2021; Anklin et al., 2021; Jaume et al., 2021). The entities can be biologically-defined, *e.g.*, nuclei, tissue regions, glands (Zhou et al., 2019; Pati et al., 2021; Anklin et al., 2021), or can be patches (Adnan et al., 2020; Aygunes et al., 2020). The entity-graphs enable to simultaneously capture local entity environment and global tissue composition. They can seamlessly scale to arbitrary tissue dimensions by incorporating arbitrary number of entities and interactions, thus offering an alternate to Multiple Instance Learning (MIL) (Campanella et al., 2019; Ming et al., 2021). The entity-graphs

also enable to selectively operate on diagnostically relevant entities, instead of analyzing the entire tissue (Tellez et al., 2019b; Shaban et al., 2020). Furthermore, when the entities depict biological units, such as nuclei, glands etc., the analysis allows pathologists to directly comprehend and reason with the outcomes (Jaume et al., 2020, 2021). However, constructing an entity-graph based pathological workflow demands several prerequisites, such as entity detection, entity encoding, constructing the graph topology etc., alongside standard preprocessing, such as stain normalization, tissue detection etc. Additionally, the workflow requires to utilize the recent advancements in DL for processing graph-structured data. All these may inhibit the adoption of entity-graphs in computational pathology. In addition, the lack of a standardized framework with the aforementioned functionalities urge the researchers to reinvent the wheel, which is cumbersome, time-consuming, hampers reproducibility, and requires a wide range of technical acumen.

To overcome these constraints, we present HISTOCARTOGRAPHY, a novel open-source python library that facilitates graph-analytics in computational pathology. Specifically our contributions are: (i) a standardized python library that unifies a set of histology image manipulation tools, entity-graph builders, GNN models, and model explainability tools, (ii) a benchmark assessment of performance and scalability on classification and segmentation tasks in pathology, (iii) a comprehensive overview of graph representation and modeling in histology, and (iv) a review of extant libraries for histological image analysis.

## 2. Related Work

### 2.1 Graphs in Computational Pathology

Entity-graphs are proposed to realize the tissue composition-to-functionality relationship in terms of the phenotypical and structural characteristics of tissue. The entities can be nuclei (Demir et al., 2004; Zhou et al., 2019; Wang et al., 2020; Chen et al., 2020; Pati et al., 2021), tissue regions (Pati et al., 2021), patches (Anand et al., 2020; Adnan et al., 2020; Aygunes et al., 2020; Zhao et al., 2020; Li et al., 2018; Levy et al., 2021), etc. Typically nodes include handcrafted or DL features to characterize the entities, and the topology can depict the spatial or semantic relationship among the entities, *e.g.*, k-Nearest Neighbors (k-NN), region adjacency, or probabilistic models. The graphs can be processed using classic Machine Learning (ML) (Sharma et al., 2016, 2017) or GNNs to outperform state-of-the-art Convolutional Neural Network (CNN)-based approaches for several pathology tasks across multiple organs (Garciá-Arteaga et al., 2017; Zhou et al., 2019; Zhao et al., 2020; Adnan et al., 2020; Pati et al., 2021; Studer et al., 2021; Anklin et al., 2021). Interestingly, when the graph-nodes depict biological entities, *e.g.*, nuclei, tissue regions, the entity-graphs combined with feature attribution techniques can provide pathologist-friendly interpretations (Zhou et al., 2019; Jaume et al., 2020; Sureka et al., 2020) and explanations (Jaume et al., 2021), unlike pixelated blurry saliency maps. A detailed review of graphs in computational pathology is presented by Ahmedt-Aristizabal et al. (2021).

### 2.2 Extant Libraries in Computational Pathology

Several open-source libraries facilitate the development of computational pathology workflows. Most of them include helper functions to perform standard preprocessing and vi-

sualization. HISTOLAB (Arbitrio et al.) includes Whole Slide Image (WSI)-level tissue detection and tile extraction modules. SYNTAX (Byfield et al., b) provides the same features with abstraction where modules can be stacked and run in a pre-defined pipeline. STAINTOOLS (Byfield et al., a) provides tools for stain normalization and augmentation. HISTOMICSTK (Beezley et al.) enables to perform tissue detection, object detection and segmentation, image filtering, stain normalization and deconvolution, and handcrafted feature extraction. Further, HISTOMICSTK allows nuclei segmentation and classification using classical ML approaches. It also provides a User Interface (UI) to run containerized modules and pipelines. Though HISTOMICSTK includes valuable functionalities, it caters limited DL tools. Similarly, OPENSLIDE (Gilbert et al.) provides a UI to read and visualize histology images that supports most of the WSI formats. Finally, QUPATH (Bankhead et al.) offers a UI that allows to read, visualize and annotate WSIs. It also includes tools to perform stain normalization, nuclei and tissue detection, and implement basic ML models. However, QUPATH is not a python Application Programming Interface (API), which makes it difficult to integrate into existing workflow and DL frameworks, *e.g.*, PyTorch, Tensorflow. Most importantly, none of the frameworks provide graph-related helpers. With the advent of graph-techniques as a new paradigm for analyzing histology images, a standardized library is desired for reinforcing the development.

## 3. Histocartography: Graph Analytics Tool for Pathology

In this section, we highlight the core functionalities of HISTOCARTOGRAPHY, (1) *Preprocessing* module: a set of histology image processing tools and entity-graph builders, (2) *ML* module: helpers to learn from entity-graphs, (3) *Explainability* module: a set of GNN model interpretability tools. The specific functionalities in each module are summarized in Table 1.

### 3.1 Preprocessing Module

**Stain normalization:** Variation in Hematoxylin and Eosin (H&E) staining protocols for tissue specimens induces appearance variability that adversely impacts computational methods (Tellez et al., 2019a). To alleviate these variations, HISTOCARTOGRAPHY implements two popular normalization algorithms proposed by Macenko et al. (2009) and Vahadane et al. (2016), similar to STAINTOOLS and HISTOMICSTK, which supports both reference-based and reference-free normalization, *i.e.*, with manual stain vectors. Figure 1 highlights a sample normalization output using our API.

**Tissue Detection:** A WSI usually includes significant non-tissue region. Identifying the tissue regions can confine the analysis and reduce computational effort. The tissue detector in HISTOCARTOGRAPHY iteratively applies Gaussian smoothing and Otsu thresholding until the mean of non-tissue pixels is below a threshold. This module is common across HISTOLAB, SYNTAX, HISTOMICSTK and QUPATH.

**Nuclei detection:** This module enables to segment and locate nuclei in H&E images. Though it is well-studied in computational pathology, only a few public implementations are available. For instance, QUPATH allows to detect nuclei but requires model training and fine-tuning. While providing flexibility, the module includes only elementary ML methods. HISTOCARTOGRAPHY integrates two checkpoints for the state-of-the-art

Table 1: Overview of HISTOCARTOGRAPHY functionalities, with the i/o, CPU and GPU compatibility, and availability in extant libraries for individual module. `I`, `M`, `X`, `G`, `P` and `S` denote an image (np.array (Harris et al., 2020)), a mask (np.array), features (torch.Tensor (Paszke et al., 2019)), a graph (DGLGraph (Wang et al., 2019)), predictions (torch.Tensor) and importance scores (torch.Tensor), respectively.

| Function | Module | Input | Output | Existing | CPU | GPU |
|---|---|---|---|---|---|---|
| Preprocessing | Vahadane Stain Norm | I | I | ✓ | ✓ | ✗ |
| | Macenko Stain Norm | I | I | ✓ | ✓ | ✗ |
| | Tissue Mask Detection | I | M | ✓ | ✓ | ✗ |
| | Nuclei Detection | I | M | ✓ | ✓ | ✓ |
| | Nuclei Concepts | I, M | M | ✓ | ✓ | ✗ |
| | Tissue Component Detection | I | M | ✗ | ✓ | ✗ |
| | Deep Feature Extraction | I, M | X | ✗ | ✓ | ✓ |
| | Feature Cube Extraction | I | X | ✗ | ✓ | ✓ |
| | k-NN Graph Building | X, M | G | ✗ | ✓ | ✗ |
| | RAG Graph Building | X, M | G | ✗ | ✓ | ✗ |
| ML | Cell-Graph Model | G | P | ✗ | ✓ | ✓ |
| | Tissue-Graph Model | G | P | ✗ | ✓ | ✓ |
| | HACT Model | G, G, X | P | ✗ | ✓ | ✓ |
| Explainers | GNNEXPLAINER | G | S | ✗ | ✓ | ✓ |
| | GRAPHGRAD-CAM | G | S | ✗ | ✓ | ✓ |
| | GRAPHGRAD-CAM++ | G | S | ✗ | ✓ | ✓ |
| | GRAPHLRP | G | S | ✗ | ✓ | ✓ |

HoVerNet model (Graham et al., 2019) trained on PanNuke (Gamper et al., 2020) and MoNuSac (Ruchika et al., 2020) datasets for nuclei segmentation and classification.

**Tissue Component Detection:** HISTOCARTOGRAPHY includes an unsupervised super-pixel based approach to segment tissue regions. First, the tissue is oversegmented into homogeneous superpixels using Simple Linear Iterative Clustering (SLIC) (Achanta et al., 2012) algorithm. Then, neighboring superpixels are hierarchically merged using color similarity to denote meaningful tissue regions, *e.g.*, epithelium and stroma regions. Superpixels depicting tissue regions are used by Bejnordi et al. (2015); Pati et al. (2020, 2021).

**Feature Extraction:** HISTOCARTOGRAPHY includes two types of feature extractors, *i.e.*, handcrafted- and CNN-based, to encode the entity characteristics. The handcrafted feature extractor computes entity-level morphological and topological properties. Morphological features capture the shape and size, *e.g.*, entity area, eccentricity, perimeter etc., and the texture captures chromaticity using the gray-level co-occurrence matrix. Topological features capture the local entity distribution using k-NN entity density estimation. A comprehensive list is provided in the Appendix. Handcrafted features can be used for training DL algorithms (Demir et al., 2004; Zhou et al., 2019; Pati et al., 2020; Studer et al., 2021), or concept-based post-hoc explainability (Jaume et al., 2021).

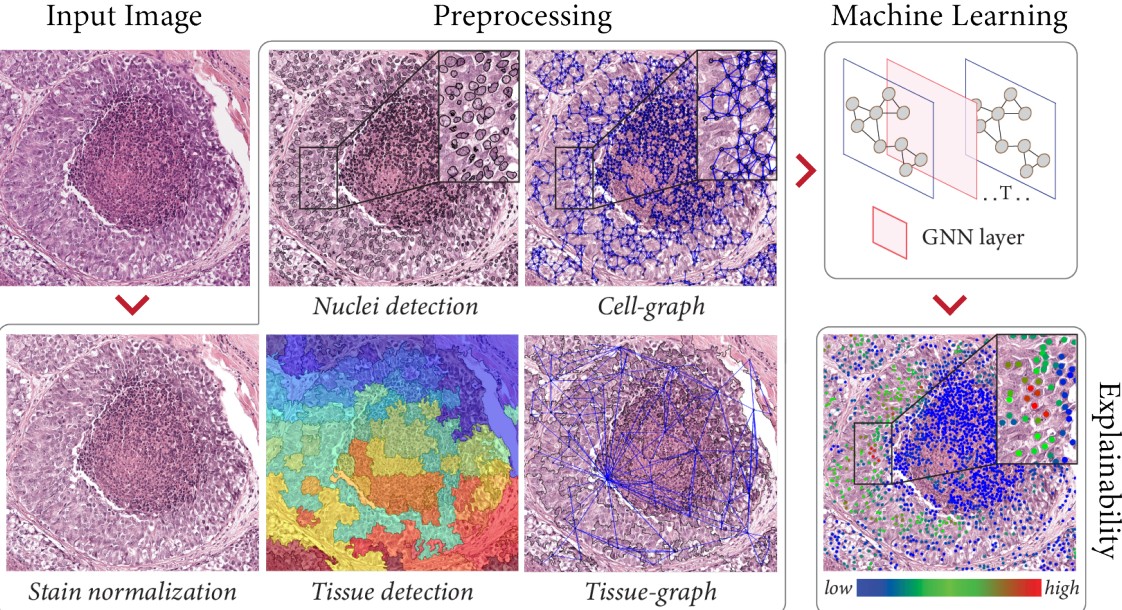

Figure 1: Overview of HISTOCARTOGRAPHY functionalities and modules.

The deep feature extractor allows to extract CNN features by using any pre-trained deep architecture, *e.g.*, ResNet, MobileNet, embedded in torchvision (Marcel et al., 2010). The module intakes patches centered around the entity and extracts features from the penultimate layer of the architectures. If the entity is larger than the specified patch size, then multiple patches within the entity, w/ or w/o overlapping, are processed, and the final feature is computed as the mean of the patch-level deep features, as used in Chen et al. (2020); Pati et al. (2020, 2021). Deep features can alternatively be extracted from the WSI to build a feature-cube as suggested by Shaban et al. (2020); Tellez et al. (2019b).

**Graph builders:** HISTOCARTOGRAPHY presents two graph builders, *i.e.*, the thresholded k-NN and the Region Adjacency Graph (RAG). The k-NN graph builder defines the graph topology by connecting each entity to its k-closest neighbors. Connections between distant entities beyond a threshold can be pruned to have spatial sparsity in the graph. We recommend this builder to connect single entities, *e.g.*, nuclei, glands. The RAG builder connects entities using spatial adjacency, *i.e.*, entities sharing a common boundary. It builds a sound topology when dealing with dense segmentation maps, *e.g.*, tissue regions. Figure 1 presents samples of cell- and tissue-graphs. Further, the module fuses the node features and the topological distribution to render a Deep Graph Library (DGL) graph for an image.

### 3.2 Graph Machine Learning Module

HISTOCARTOGRAPHY includes a set of DL models, based on a GNN backbone to learn from graph-structured tissue representations. It includes two state-of-the-art GNN layers, *i.e.*, Graph Isomorphism Network (GIN) (Xu et al., 2019) and Principal Neighboorhood Aggregation (PNA) (Corso et al., 2020). PNA proves to outperform GIN provided more computational resources (Dwivedi et al., 2020). HISTOCARTOGRAPHY defines cell- and tissue-graph models, which are GNN-based abstractions to learn from biological entity-graphs. They

offer efficient (Pati et al., 2021), scalable (Anklin et al., 2021) and explainable (Jaume et al., 2020, 2021) approaches to analyze histology images. Further, the library includes models to jointly represent and learn from cell- and tissue-graphs (Pati et al., 2021). The models in HISTOCARTOGRAPHY are organized such that they can be adapted to various GNN backbones, tasks (*e.g.*, regression, clustering, classification, segmentation), organs, and entity-types. These models provide the blueprints to accelerate the development of graph-based models in computational pathology. All the graph modules are implemented using DGL (Wang et al., 2019), a state-of-the-art library for GNNs built around PyTorch.

### 3.3 Explainability Module

HISTOCARTOGRAPHY includes four post-hoc feature attribution graph explainers, that can generate node-level saliency maps to highlight the node-wise contribution towards an output task. Namely, the library includes two gradient-based explainers (GRAPHGRAD-CAM (Selvaraju et al., 2017; Pope et al., 2019) and GRAPHGRAD-CAM++ (Chattopadhay et al., 2018; Jaume et al., 2021)), a node pruning-based explainer (GNNEXPLAINER (Ying et al., 2019)), and a layer-wise relevance propagation explainer (GRAPHLRP (Schwarzenberg et al., 2019)). The saliency map can be visualized by overlaying the node importances on the input image (see Figure 2). Alternatively, entities with high importances can be extracted and studied independently to assess their relevance (Jaume et al., 2021).

### 3.4 Pipeline Runner

To facilitate an easy-to-use and human-readable development, HISTOCARTOGRAPHY includes a pipeline runner. It allows to define a series of pipeline steps along with loading and saving utilities to reduce boilerplate code.

## 4. Benchmarking HistoCartography

We benchmark HISTOCARTOGRAPHY in terms of run-time and performance for various histopathology tasks, *i.e.*, stain normalization, tissue detection, tumor classification and segmentation etc., on images of varying dimensions. The CPU and GPU compatible modules are assessed on a single-core POWER8 processor and a NVIDIA P100 GPU, respectively.

### 4.1 Computational Time

Analyzing the computational time for processing a histology image is imperative. We thoroughly analyze the run-time of HISTOCARTOGRAPHY modules on a set of Regions of Interest (RoIs) and WSIs. The analyses are presented in Table 2, and a comprehensive extension is provided in Supplementary: Table 1. The preprocessing modules are observed to be the most time-consuming. For instance, Vahadane stain normalization can take up to 3 minutes to process a $11'000 \times 11'000$ image, whereas Macenko method is $2\times$ faster for competitive result. The implementations are computationally similar to HISTOLAB and STAINTOOLS, and scale linearly w.r.to image size. The cell- and tissue-graph construction take 2.5 and 4.1 seconds respectively for a $1000 \times 1000$ image with the following parameters. Nuclei detection is performed on patches of size $256 \times 256$ with an overlap of 164 pixels. Nuclei features are extracted from $72 \times 72$ patches centered around the nuclei, that are resized to

Table 2: Run time analysis of HISTOCARTOGRAPHY core functionalities (in seconds).

| Functionalities | Modules | Tumor RoI | | | WSI | | |
|---|---|---|---|---|---|---|---|
| | Size | $1000^2$ | $2500^2$ | $5000^2$ | $5000^2$ | $7500^2$ | $11000^2$ |
| Preprocessing | Vahadane Stain Norm | 1.77 | 6.46 | 29.03 | 30.67 | 68.27 | 186.10 |
| | Macenko Stain Norm | 0.80 | 2.86 | 11.19 | 15.98 | 32.37 | 81.72 |
| | Tissue Mask Detection | - | - | - | 1.04 | 2.11 | 8.09 |
| | Feature Cube Extraction | 0.24 | 1.61 | 5.92 | 5.83 | 11.77 | 30.21 |
| | Cell-Graph Generation | 2.51 | 13.33 | 50.26 | - | - | - |
| | Tissue-Graph Generation | 4.14 | 21.30 | 83.66 | 39.18 | 93.26 | 276.71 |
| ML | Cell-Graph Model | 0.028 | 0.033 | 0.040 | - | - | - |
| | Tissue-Graph Model | 0.011 | 0.015 | 0.026 | 0.039 | 0.056 | 0.069 |
| | Hierarchical Model | 0.034 | 0.041 | 0.057 | - | - | - |
| Explainers | GNNEXPLAINER | 12.00 | 13.09 | 35.33 | - | - | - |
| | GRAPHGRAD-CAM | 0.011 | 0.022 | 0.035 | 0.025 | 0.030 | 0.033 |
| | GRAPHGRAD-CAM++ | 0.011 | 0.023 | 0.035 | 0.026 | 0.030 | 0.033 |
| | GRAPHLRP | 0.020 | 0.024 | 0.90 | 0.079 | 0.085 | 0.089 |

$224 \times 224$ and processed by ResNet34 pretrained on ImageNet (Deng et al., 2009). Finally, thresholded k-NN topology is built with $k = 5$ and a threshold distance of 50 pixels. For the tissue-graph, SLIC is used to extract 400 superpixels per image, that are subsequently merged to provide the tissue components. Tissue features are also extracted using ResNet34 with $144 \times 144$ size patches that are resized to $224 \times 224$. The graph buildings can be further optimized as per the task by downsampling the input image, reducing the patch overlap, or by using a lighter feature extractor. For extracting the feature cube representation, we process patches of size $144 \times 144$ resized to $224 \times 224$ w/o overlap by pretrained ResNet34.

Tumor RoIs (TRoIs) are processed using a cell- and tissue-graph model, and the hierarchical cell-to-tissue graph model (Pati et al., 2021). They consist of three PNA layers with 64 hidden units followed by an additional 2-layer MLP with 128 hidden units for classification. WSIs are processed using SEGGINI Anklin et al. (2021), a weakly supervised approach basdn on tissue-graphs, which contains six GIN layers with 64 hidden units followed by a 2-layer MLP with 128 hidden units. The models process in near real-time irrespective of the increment in the graph size. The graph explainers are based on GNNs with 3 GIN layers, each having a 2-layer MLP with 32 hidden units, and a 2-layer MLP head. GNNEXPLAINER is the slowest among all as it requires to optimize a mask to explain each image.

## 4.2 Performance Benchmark

Table 3 benchmarks the performance of HISTOCARTOGRAPHY for classification and segmentation tasks. Classification is performed on BRACS (Pati et al., 2021) and BACH (Aresta et al., 2019) datasets to characterize breast tumors using cell-graph model, tissue-graph model, and HACT-Net (Pati et al., 2021), and the performance is measures by weighted-F1 score. Segmentation is performed using SEGGINI (Anklin et al., 2021) to delineate Gleason patterns in prostate cancer images from UZH (Zhong et al., 2017) and SICAPv2 (Silva-Rodríguez et al., 2020), and the performance is measured by average Dice score. We evaluate

Table 3: Benchmarking HISTOCARTOGRAPHY for classification and segmentation (in %).

| Task | Dataset | Model | Image Type | Avg. #pixels | #classes | Avg. Dice | Weighted F1 |
|---|---|---|---|---|---|---|---|
| Classification | BRACS | CG-GNN | TRoI | $3.9 \times 10^6$ (40×) | 7 | - | $55.9 \pm 1.0$ |
| | BRACS | TG-GNN | TRoI | $3.9 \times 10^6$ (40×) | 7 | - | $56.6 \pm 1.3$ |
| | BRACS | HACT-Net | TRoI | $3.9 \times 10^6$ (40×) | 7 | - | $61.5 \pm 0.9$ |
| | BACH | HACT-Net | TRoI | $3.1 \times 10^6$ (20×) | 4 | - | $90.7 \pm 0.5$ |
| | SICAPv2 | SEGGINI | WSI | $121 \times 10^6$ (10×) | 6 | - | $62.0 \pm 3.6$ |
| | UZH | SEGGINI | TMA | $9.6 \times 10^6$ (40×) | 6 | - | $56.8 \pm 1.7$ |
| Seg. | SICAPv2 | SEGGINI | WSI | $121 \times 10^6$ (10×) | 4 | $44.3 \pm 2.0$ | - |
| | UZH | SEGGINI | TMA | $9.6 \times 10^6$ (40×) | 4 | $66.0 \pm 3.1$ | - |

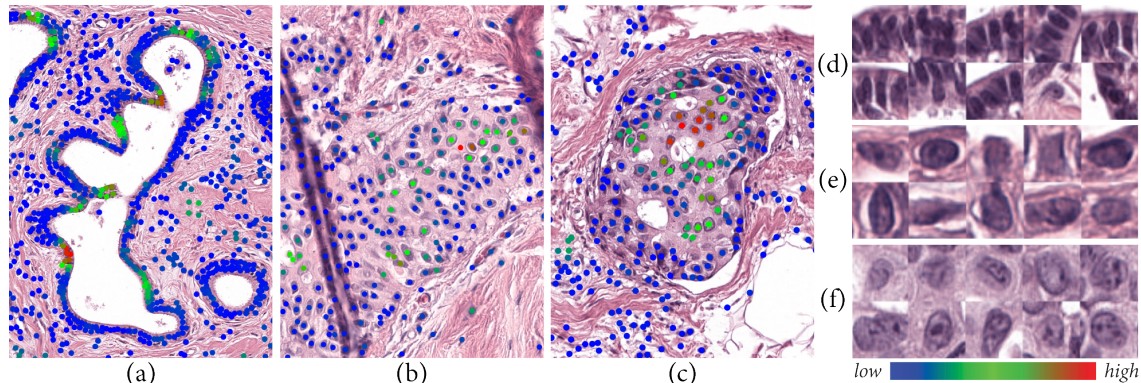

(a)          (b)          (c)

(d)
(e)
(f)
*low* ▆▆▆▆ *high*

Figure 2: Qualitative explanations of sample breast RoIs: (a) Benign, (b) ADH, (c) DCIS.
(d, e, f) highlight the ten most important nuclei for the respective samples.

on various image types, *i.e.*, tumor RoIs, tissue microarrays, and whole-slides, to highlight the scalability of entity-graphs in HISTOCARTOGRAPHY to arbitrary image dimensions.

### 4.3 Qualitative Explanations

Figure 2 presents the outcome of GRAPHGRADCAM module in HISTOCARTOGRAPHY to interpret a cell-graph model. This module renders per-image explanations in terms of node-level saliency maps by applying post-hoc feature attribution methods on trained cell-graph model. Further, the cell-graph model can be interpreted by characterizing the highlighted important nuclei per-image, as shown in Figure 2.

### 5. Conclusion

We introduced HISTOCARTOGRAPHY, the first open source library, to the best of our knowledge, to facilitate graph analytics, *i.e.*, graph representation, learning, and explainability, in computational pathology. It can potentially enable researchers to develop entity-graph based pathology workflows by leveraging the inbuilt helpers. As the library is built on python, the deep learning researchers can seamlessly customize and integrate the functionalities into their task-specific workflows. HISTOCARTOGRAPHY is constantly growing with new functionalities and improved implementations, aiming to promote the adoption of graph-based analysis in computational pathology.

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

## Supplementary

### HistoCartography Syntax

In this section, we introduce the syntax to implement the functionalities of HISTOCARTOG-RAPHY. Figure 1 presents code snippets to implement Vahadane stain normalization and tissue mask detection. Figure 2 shows the syntax for building cell- and tissue-graphs. Noticeably, these functionalities require only ten lines of code by using HISTOCARTOGRAPHY, which could have otherwise required a few hundred lines. In Figure 3, we present the syntax to declare and run a cell- and tissue-graph model. All the model parameters, *e.g.*, GNN type, number of GNN layers, can be adapted and fine-tuned using a configuration file. Finally, Figure 4 shows code snippets to use the graph explainability modules. All explainers follow a similar syntax with the same input and output types, making implementation and integration straightforward.

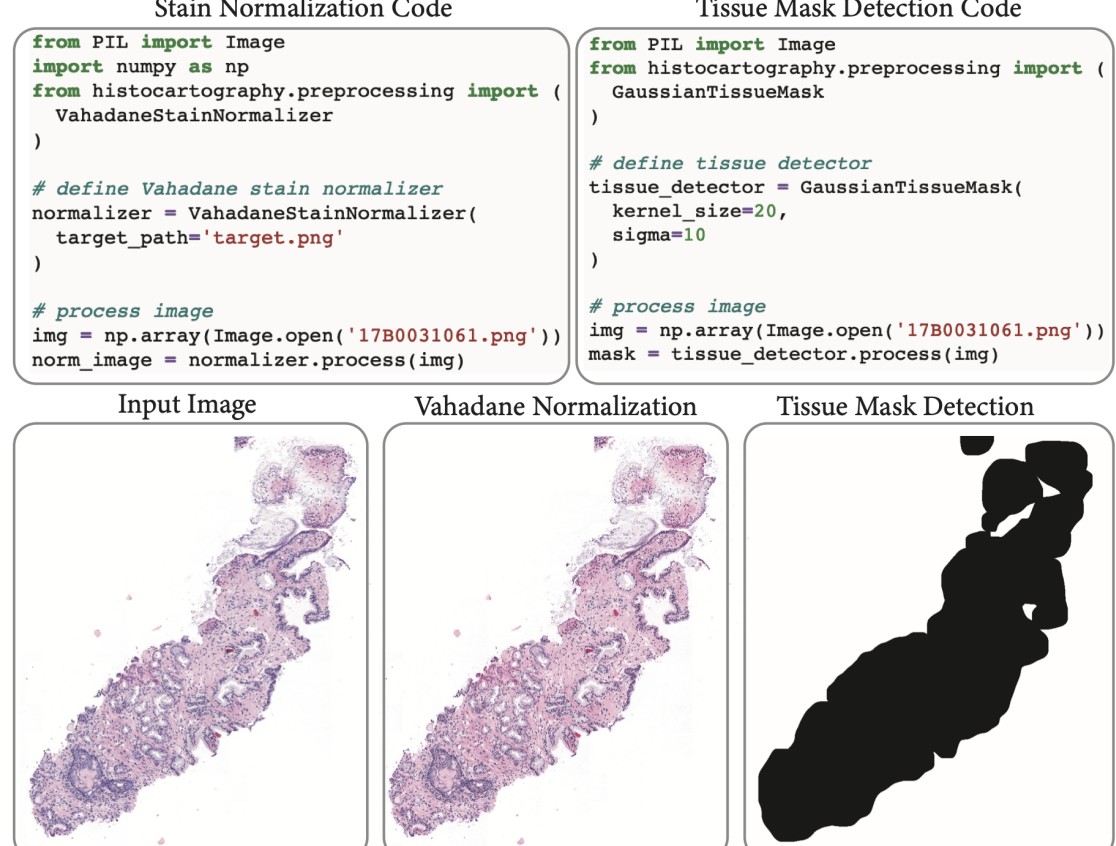

Figure 1: Implementation of Vahadane stain normalization (left) and tissue mask detection (right) with the *Preprocessing* functionalities in the HISTOCARTOGRAPHY API.

**Cell Graph Generation Code**

```python
from PIL import Image
from histocartography.preprocessing import (
  NucleiExtractor,
  DeepFeatureExtractor,
  KNNGraphBuilder
)

# define nuclei
nuclei_detector = NucleiExtractor()

# define feature extractor
feats_extractor = DeepFeatureExtractor(
    architecture='resnet34',
    patch_size=72,
    resize_size=224
)

# define graph builder
knn_graph_builder = KNNGraphBuilder(
  k=5,
  thresh=50
)

# process image
img = np.array(Image.open('image.png'))
nuclei, _ = nuclei_detector.process(img)
feats = feats_extractor.process(img, nuclei)
graph = knn_graph_builder.process(nuclei, feats)
```

**Tissue Graph Generation Code**

```python
from PIL import Image
from histocartography.preprocessing import (
    ColorMergedSuperpixelExtractor,
    DeepFeatureExtractor,
    RAGGraphBuilder
)

# define super-pixels
superpx_detector = ColorMergedSuperpixelExtractor(
    nr_superpixels=400,
    downsampling_factor=4
)

# define feature extractor
feats_extractor = DeepFeatureExtractor(
    patch_size=144,
    resize_size=224
)

# define graph builder
rag_graph_builder = RAGGraphBuilder()

# process image
img = np.array(Image.open('image.png'))
superpxs, _ = superpx_detector.process(img)
feats = feats_extractor.process(img, superpxs)
graph = rag_graph_builder.process(superpxs, feats)
```

Input Image      Cell Graph      Tissue Graph

Figure 2: Implementation of cell-graph (left) and tissue-graph (right) generation using the graph builders in HISTOCARTOGRAPHY.

**Handcrafted Feature Extraction**

In this section, we provide a comprehensive list of morphological and topological features which can be extracted per-entity by HISTOCARTOGRAPHY. Morphological features include shape, size and texture properties, namely, entity area, convex area, eccentricity, equivalent diameter, euler number, length of the major and minor axis, orientation, perimeter, solidity, convex hull perimeter, roughness, shape factor, ellipticity, roudness. Texture properties are based on gray-level co-occurrence matrices (GLCM). Specifically, we extract the GLCM contrast, dissimilarity, homogeneity, energy, angular speed moment and dispersion. The topological features are based on the entity density computed as the mean and variance of entity crowdedness. These features can be computed for the most important set of entities highlighted by the graph explainability techniques, and utilized along with prior pathological knowledge to interpret the trained entity-graph models.

Cell Graph Model

```python
import yaml
from dgl.data.utils import (
    load_graphs
)
from histocartography.ml import (
    CellGraphModel
)

# load model configurations
cfg = yaml.safe_load(open('cg_cfg.yml', 'r'))

# define cell graph model
model = CellGraphModel(
    gnn_params=cfg['gnn_params'],
    classification_params=cfg['cls_params'],
    node_dim=512,
    num_classes=3
)

# load cell graph
cg, _ = load_graphs('cg.bin')

# forward pass
logits = model(cg)
```

Tissue Graph Model

```python
import yaml
from dgl.data.utils import (
    load_graphs
)
from histocartography.ml import (
    TissueGraphModel
)

# load model configurations
cfg = yaml.safe_load(open('tg_cfg.yml', 'r'))

# define tissue graph model
model = TissueGraphModel(
    gnn_params=cfg['gnn_params'],
    classification_params=cfg['cls_params'],
    node_dim=512,
    num_classes=3
)

# load tissue graph
tg, _ = load_graphs('tg.bin')

# forward pass
logits = model(tg)
```

Figure 3: Implementation of the cell- (left) and tissue- graph (right) model by using the ML modules in the HISTOCARTOGRAPHY API

Cell Graph Explainer Code

```python
from histocartography.interpretability import (
    GraphGradCAMExplainer,
    GraphGradCAMPPExplainer,
    GraphPruningExplainer,
    GraphLRPExplainer
)

# load pretrained model
model = CellGraphModel(config['gnn_params'], config['cls_params'], 512, pretrained=True)

# load cell graph
graph, _ = load_graphs('291_dcis_18.bin')

# define graph explainers
grad_cam_explainer = GraphGradCAMExplainer(model=model)
grad_campp_explainer = GraphGradCAMPPExplainer(model=model)
gnn_explainer = GraphPruningExplainer(model=model)
graph_lrp_explainer = GraphLRPExplainer(model=model)

# explain cell graph
grad_cam_scores, _ = grad_cam_explainer.process(graph)
grad_campp_scores, _ = grad_campp_explainer.process(graph)
gnn_explainer_scores, _ = gnn_explainer.process(graph)
graph_lrp_scores, _ = graph_lrp_explainer.process(graph)
```

GNNExplainer      GraphGradCAM      GraphGradCAM++      GraphLRP

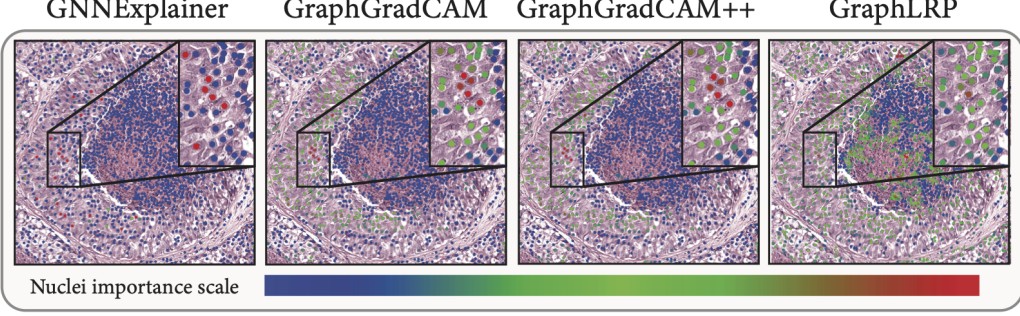

Nuclei importance scale

Figure 4: Implementation of graph explainers in HISTOCARTOGRAPHY. The most important nodes are marked in red and the least important ones in blue.

Table 1: Extended version of Table 2. Reported time to run HISTOCARTOGRAPHY core functionalities. CPU-only experiments were run on a single-core POWER8 processor, and GPU-compatible experiments were run on an NVIDIA P100 GPU. Time is reported in seconds.

| | | Modality | Tumor RoI | | | WSI | | |
|---|---|---|---|---|---|---|---|---|
| | | Size | $1000^2$ | $2500^2$ | $5000^2$ | $5000^2$ | $7500^2$ | $11000^2$ |
| Preprocessing | Standard | Vahadane Normalization | 1.77 | 6.46 | 29.03 | 30.67 | 68.27 | 186.10 |
| | | Macenko Normalization | 0.80 | 2.86 | 11.19 | 15.98 | 32.37 | 81.72 |
| | | Tissue Mast Detection | - | - | - | 1.04 | 2.11 | 8.09 |
| | | Feature Cube Extraction | 0.24 | 1.61 | 5.92 | 6.27 | 11.97 | 29.79 |
| | CG | Nuclei Detection | 3.03 | 12.93 | 47.66 | - | - | - |
| | | Nuclei Concept Extraction | 2.95 | 6.52 | 27.94 | - | - | - |
| | | Deep Nuclei Feature Extraction | 0.10 | 0.30 | 1.28 | - | - | - |
| | | k-NN Graph Building | 0.06 | 0.20 | 1.35 | - | - | - |
| | TG | Super-pixel Detection | 3.32 | 17.84 | 68.99 | 31.50 | 68.99 | 183.54 |
| | | Deep Tissue Feature Extraction | 0.56 | 2.99 | 8.40 | 4.17 | 9.96 | 20.54 |
| | | RAG Graph Building | 0.12 | 2.04 | 25.6 | 6.33 | 19.98 | 85.73 |
| ML | | Cell-Graph Model | 0.028 | 0.033 | 0.040 | - | - | - |
| | | Tissue-Graph Model | 0.011 | 0.015 | 0.026 | 0.039 | 0.056 | 0.069 |
| | | HACT Model | 0.034 | 0.041 | 0.057 | - | - | - |
| Explainers | CG | GNNEXPLAINER | 12.00 | 13.09 | 35.33 | - | - | - |
| | | GRAPHGRAD-CAM | 0.011 | 0.022 | 0.035 | - | - | - |
| | | GRAPHGRAD-CAM++ | 0.011 | 0.023 | 0.035 | - | - | - |
| | | GRAPHLRP | 0.020 | 0.024 | 0.90 | - | - | - |
| | TG | GNNEXPLAINER | 11.23 | 11.28 | 11.38 | - | - | - |
| | | GRAPHGRAD-CAM | 0.011 | 0.012 | 0.018 | 0.025 | 0.030 | 0.033 |
| | | GRAPHGRAD-CAM++ | 0.011 | 0.013 | 0.018 | 0.026 | 0.030 | 0.033 |
| | | GRAPHLRP | 0.011 | 0.014 | 0.016 | 0.079 | 0.085 | 0.089 |

