# OpenReview forum: "HistoCartography: A Toolkit for Graph Analytics in Digital Pathology"
_MICCAI.org/2021/Workshop/COMPAY — COMPAY 2021_

### Official Review · Reviewer_mGdK · 2021-08-22
**HistoCartography**

**Rating:** 7
**Confidence:** 5

**Review:**

The authors present a python API for the analysis of digital pathology images using graph-based techniques. Overall, the paper is well written and the goals of the paper are made clear from the start.

The python API will be a benefit to the computational pathology (CPath) community and can help speed up experimentation for graph-based approaches in CPath. A few points:
- Why is the WSI time not given for the cell-based models? It needs to be made clear whether the API supports both image regions and WSIs as input.
- How extendable is the code? Can a user modify the current implementation for their purpose? For example, a limited number of GNNs are supplied as part of the API, but maybe the user wants to add a different GNN from the DGL library. This also applies to the deep feature extractor. Maybe a user may want to use their own deep network to extract instance-based features. Please discuss this.
- Is testing & code coverage implemented in the library? This will ensure that the code is reliable.
- Are examples given / provided in the library? Jupyter notebooks would be beneficial.
- If the paper is accepted, please add the relevant links to access the python code and make sure that the repository / library is referenced more often.

Use the below reference, rather than the researchgate one:
Verma, Ruchika, et al. "MoNuSAC2020: A Multi-organ Nuclei Segmentation and Classification Challenge." IEEE Transactions on Medical Imaging (2021).

---

### Official Review · Reviewer_BY8T · 2021-08-23
**A Graph Analytics Tool is presented for pathology and the core functionalities of this tool is explained very well, along with some examples of using this tool.**

**Rating:** 9
**Confidence:** 5

**Review:**

This paper is very clearly written along with useful examples. This tool has the potential to grow more in future and embed more features and processes that can be used in histopathology analysis.

The only comment I put is that the citing format is different throughout the manuscript. The authors could use the (author, year) format.

---

### Decision · Program_Chairs · 2021-08-25

Accept